# Age-Associated *TET2* Mutations: Common Drivers of Myeloid Dysfunction, Cancer and Cardiovascular Disease

**DOI:** 10.3390/ijms21020626

**Published:** 2020-01-17

**Authors:** Christina K. Ferrone, Mackenzie Blydt-Hansen, Michael J. Rauh

**Affiliations:** Department of Pathology and Molecular Medicine, Queen’s University, Kingston, ON K7L 3N6, Canada; christina.ferrone@queensu.ca (C.K.F.); 15mbh4@queensu.ca (M.B.-H.)

**Keywords:** clonal hematopoiesis, *TET2*, driver mutations, NGS, clinical detection, inflammation, cancer progression, comorbid disease, aging, targeting TET2 therapeutically

## Abstract

Acquired, inactivating mutations in Tet methylcytosine dioxygenase 2 (*TET2*) are detected in peripheral blood cells of a remarkable 5%–10% of adults greater than 65 years of age. They impart a hematopoietic stem cell advantage and resultant clonal hematopoiesis of indeterminate potential (CHIP) with skewed myelomonocytic differentiation. CHIP is associated with an overall increased risk of transformation to a hematological malignancy, especially myeloproliferative and myelodysplastic neoplasms (MPN, MDS) and acute myeloid leukemia (AML), of approximately 0.5% to 1% per year. However, it is becoming increasingly possible to identify individuals at greatest risk, based on CHIP mutational characteristics. CHIP, and particularly *TET2*-mutant CHIP, is also a novel, significant risk factor for cardiovascular diseases, related in part to hyper-inflammatory, progeny macrophages carrying *TET2* mutations. Therefore, somatic *TET2* mutations contribute to myeloid expansion and innate immune dysregulation with age and contribute to prevalent diseases in the developed world—cancer and cardiovascular disease. Herein, we describe the impact of detecting *TET2* mutations in the clinical setting. We also present the rationale and promise for targeting *TET2*-mutant and other CHIP clones, and their inflammatory environment, as potential means of lessening risk of myeloid cancer development and dampening CHIP-comorbid inflammatory diseases.

## 1. Introduction

Recent technological advancements have permitted the identification of a myriad of genes mutated in myeloid cancers [1,2,3,4,5,6,7]. *TET2* is one of the most commonly mutated genes in these diseases, being highly involved in epigenetic regulation, including cytosine demethylation [5,8]. *TET2*-inactivating mutations also frequently occur in clonal hematopoiesis of indeterminate potential (CHIP), a “pre-leukemic” condition involving aberrant clonal expansion of hematopoietic stem and progenitor cells (HSPCs) in the bone marrow [9,10]. Herein, we explore the importance of detecting mutations such as those observed in *TET2* and its partner DNA-methyltransferase 3A (*DNMT3A*; another key epigenetic regulator, frequently mutated in CHIP and myeloid disease), and some of the associations of these genetic variations with CHIP and myeloid disease progression. We further discuss recent findings regarding CHIP, *TET2* mutations and their connections to increased inflammation and comorbid disease. Finally, we briefly describe recent attempts to therapeutically increase wildtype TET2 protein levels or activity in the presence of heterozygous loss-of-function mutations.

## 2. Hematopoiesis and Myeloid Malignancies

Hematopoiesis is the production of mature blood cells from HSPCs. This process is polyclonal, with a range of approximately 50,000–200,000 HSPCs each giving rise to a fraction of mature blood cells [11]. Myeloid malignancies occur when individual HSPC mutant clones proliferate, expand and skew towards myeloid lineages. These include myelodysplastic syndromes (MDS), myeloproliferative neoplasms (MPN), myelodysplastic/myeloproliferative neoplasms (MDS/MPN), and acute myeloid leukemia (AML) (Figure 1) [1]. The WHO describes MDS as having “ineffective hematopoiesis”, characterized by abnormal hematopoietic cell shapes (morphological dysplasia) and peripheral cytopenia (low blood cell counts) [1]. MPN are characterized by the proliferation of mature myeloid cells, usually lacking morphological dysplasia, while myeloid neoplasms with clinical, laboratory, and morphologic features overlapping between MDS and MPN are classified as MDS/MPN [12]. AML involves the proliferation of immature myeloid cells (“blasts”) related to mutations that block normal HSPC differentiation and, like MPN, can involve recurrent mutations associated with cellular proliferation [1,12].

## 3. Unexplained Cytopenias and Shortcomings of Current Diagnostic Techniques

Unexplained blood cytopenia is another condition, in which individuals are deficient in one or more types of blood cell (red cells, white cells, or platelets), where the origin of this deficiency is not attributable to any identifiable cause or associated disease [16]. The current diagnostic approach for MDS relies primarily on morphological studies of peripheral blood and bone marrow aspirates to identify dysplasia, and on the presence of an abnormal karyotype [17]; however, many patients with unexplained cytopenias lack characteristic features of MDS and/or display normal karyotypes. Thus, it is difficult to reach a definitive diagnosis of MDS in patients with cytopenias due to the reliance on subjective morphological assessment. The classification of idiopathic cytopenia of undetermined significance (ICUS) was adopted to define a condition of unexplained blood cytopenia that doesn’t coincide with the diagnostic criteria of MDS [18]. Conversely, clonal cytopenia of undetermined significance (CCUS) is a new term used to classify patients with ICUS that are found to possess one or more somatic mutations [19]. This classification was introduced because recent studies have shown that many patients diagnosed with ICUS also possess MDS-associated somatic mutations, in addition to potentially sharing clinical and genetic characteristics with MDS patients (Figure 1) [19,20]. Given the subjective nature of morphological assessment, more accurate and precise diagnostic tools are necessary for early stage and existing myeloid neoplasms. 

## 4. Promise of NGS in Myeloid Malignancy Diagnosis

Next-generation sequencing (NGS) has become increasingly important in the detection of genetic variants in myeloid malignancies [3,21,22]. Hematological variants identified through NGS impact diagnosis, in addition to revealing new prognostic, predictive, and therapeutic biomarkers in myeloid malignancies [21,22,23]. Several genes have been identified as recurrently mutated in these disorders, including epigenetic regulators (*TET2*, *ASXL1*, *EZH2*, *DNMT3A*, *IDH1*/*IDH2*), splicing factors (*SF3B1*, *U2AF1*, *ZRSR2*), genes involved in signal transduction (*JAK2*, *KRAS*, *NRAS*, *CBL*), among others [5,24,25]. For example, mutations in *CALR* or *JAK2* are indicative of MPN [26], while *FLT3*, *NPM1*, and *CEBPA* are associated with AML [27]. Such mutations can also be prognostic indicators; e.g., the presence of *TP53* or *FLT3* mutations are associated with worse outcomes in AML [28,29]. Recent updates to the WHO classification of myeloid neoplasms have also begun to incorporate such mutations into diagnostic criteria. Notably, the previous MDS diagnosis of refractory anemia with ring sideroblasts (RARS) required a minimum of 15% ring sideroblasts to confirm a RARS diagnosis [12], while the updated diagnosis of MDS with ring sideroblasts (MDS-RS) requires only 5% ring sideroblasts when accompanied by an *SF3B1* mutation (associated with mis-splicing of a mitochondrial iron transporter leading to abnormal accumulation of iron in mitochondria ringing the nucleus) [1]. 

## 5. CHIP and Risk of Hematological Malignancy

Hematopoietic cells tend to accumulate somatic mutations over time. It has been estimated that there are normally 50,000–200,000 HSPCs, and that they have a fidelity rate of approximately 0.78 × 10^−9^ mutations per genomic base pair per cell division, resulting in random mutations occurring at a rate of approximately 14 base substitutions and 0.13 coding mutations per year of life (i.e., approx. one mutation every 7–8 years) [7,11,30]. The prevalence of such mutations thus increases with age, but they generally do not impact normal HSPC function. However, some of these mutations can occur in genes that predispose them to malignant disease [7]. CHIP is a term used to describe individuals with hematological malignancy-associated somatic mutation(s) in the absence of other hematological malignancy diagnostic criteria [13,31]. Due to this accumulation of mutations over time, CHIP is especially common in the elderly population (>10% in those over 65 years of age), and therefore often referred to as age-related clonal hematopoiesis (ARCH) [9,13,32]. The minimum variant allele frequency (VAF) for genetic variants in individuals to meet the criteria for CHIP is ≥2% [13]. Furthermore, while CHIP is a clinical entity with specific criteria, including mutations in recognized cancer driver genes like *TET2* and *DNMT3A*, the more general term clonal hematopoiesis (CH) describes a state in which a single hematopoietic stem cell clone gives rise to a disproportionate number of an individual’s mature blood cells. CH can include mutations at VAF <2% but may also arise without recognizable cancer driver mutations, possibly resulting from contraction and neutral drift in the HSPC pool [33].

Furthermore, the presence of such mutations is associated with an increased risk of progression to myeloid malignancies [9]. Increasing clonality of hematopoietic cells resulting from an accumulation of somatic mutations can lead to disease progression, where CHIP progresses to MDS for example, which can in turn transform to AML (associated with a poorer prognosis). CHIP is associated with an overall increased risk of progression to a hematological malignancy of approximately 0.5% to 1% per year [13]. Commonalities in mutational profiles suggest that blood cancers arise from earlier clones [32]. Importantly, the introduction of NGS technologies permitted the identification of these mutations in individuals without malignant disease, including mutations in *TET2* (one of the most commonly mutated genes) [9,10].

## 6. *TET2* and *DNMT3A* Mutations and Their Impact

*TET2* is widely affected by mutations in myeloid neoplasms, and is one of the most commonly mutated genes in CHIP [9,32,34]. Somatic *TET2* mutations are present in approximately 50% of chronic myelomonocytic leukemia (CMML; an MDS/MPN) cases, ~30% of MDS, and ~10% of AML [8]. Such *TET2* loss-of-function mutations are associated with a DNA hypermethylation phenotype, tumour progression, and poor patient outcome [35,36]. DNA methylation plays a key role in proper HSPC self-renewal and lineage differentiation, and its dysregulation can lead to aberrant stem cell function and cellular transformation [37]. TET2 plays a crucial role in epigenetic modulation by promoting DNA demethylation [38]. DNA hypermethylation resulting from *TET2* mutation is associated with CHIP, increased risk of MDS progression, and poor prognosis in AML [35]. The TET2 protein itself is one of the ten-eleven-translocation (TET1-3) proteins, which are alpha-ketoglutarate- and Fe^2+^-dependent dioxygenases (α-KGDDs). These α-KGDDs primarily catalyze the oxidation of 5-methylcytosine (5mC) to 5-hydroxymethylcytosine (5hmC) (Figure 2). These represent key intermediates in DNA demethylation through replication-dependent dilution or base excision repair [39]. 

Several studies have assessed the structure of the TET2 protein and the context of somatic mutations in myeloid malignancies, including the domains in which they occur. Whether mutations occur prior to or within the C-terminal catalytic domain of TET2, they generally result in loss of enzymatic function. Studies have shown that most commonly, nonsense or frameshift mutations occur before (and occasionally within) the catalytic domain, while missense mutations and in-frame deletions occur within the catalytic domain [8,40,41] (Figure 3). To our knowledge, a comprehensive comparison of *TET2* mutations in CHIP versus myeloid malignancies has yet to be performed; however, the nature of *TET2* CHIP mutations are also in keeping with TET2 catalytic inactivation. Overall, somatic mutations in *TET2* have been found to occur in approximately 15% of patients with myeloid malignancies [8]. This gene therefore represents a promising molecular feature that can impact the clinical care of these patients. For example, the presence or absence of *TET2* can be indicative of treatment outcome [42].

In 2014, Bejar et al. found that response to hypomethylating agents (HMA) in MDS patients is enhanced in those that exhibit clonal *TET2* mutations [43]. They also found that the presence of additional mutations such as in *ASXL1* indicate poorer prognosis [43], while another group found that overall survival is greater in patients with clonal *TET2* mutations in the absence of *ASXL1* mutations [44]. However, success of HMA treatment in MDS patients has been found to be inconsistent and dependent on particular combinations of mutations, though MDS patients are not currently denied HMA therapy based on the presence of certain mutations [45,46]. Additionally, the presence of *TET2* mutations has been associated with reduced overall survival in patients with intermediate-risk AML [21]. Other studies have shown that co-mutations in *DNMT3A* and *TET2* increase the risk of malignant transformation compared to either mutation alone [47,48], which is concerning given that these genes are frequently co-mutated in MDS [3]. 

The *DNMT3A* gene is affected by loss-of-function mutations in myeloid malignancies. It is an epigenetic regulator, where its associated protein, DNMT3A, catalyzes the methylation of CpG dinucleotides in genomic DNA [49]. Specifically, it catalyzes the covalent addition of a methyl group to the C5 position of cytosine (generating 5mC) [49,50]. DNMT3A is important in HSPC differentiation and helps regulate the function of stem cells. Loss-of-function in murine *Dnmt3a* leads to HSPC expansion, clonal dominance, aberrant DNA methylation, and eventually hematological malignancies [51]. The presence of a *DNMT3A* mutation in myeloid malignancies has also been shown to predict a higher likelihood of responding to an HMA [52]. Furthermore, *DNMT3A* mutations are associated with a worse prognosis in MDS when they occur in the presence of some other mutations, especially mutations in *SF3B1* [53,54,55]. Mutations in *DNMT3A* also predict poor prognosis in AML [55].

Overall, the use of NGS to detect *TET2*, *DNMT3A*, and other mutations in patients with myeloid malignancies is imperative. These molecular features can be used as prognostic and diagnostic indicators for these patients, in addition to informing clinicians regarding the probability of disease progression and treatment options to improve patient care.

## 7. CHIP and Comorbid Diseases

While *TET2* and *DNMT3A* mutations are two of most common genetic variations in CHIP and can precede myeloid malignancies, recent studies have also revealed associations between CHIP and other comorbid diseases [9,32]. Such studies have approached the identification of CHIP-associated diseases by genetically sequencing populations of elderly individuals who are more likely to harbour pathogenic somatic variants for CHIP, revealing that somatic changes in specific genes (especially *TET2* and *DNMT3A*) are associated with increased risk of developing non-neoplastic diseases [9,32,33,56,57]. In a study in *The New England Journal of Medicine* in 2014, Jaiswal et al. observed an increase in all-cause mortality for individuals with CHIP, as well as a modestly elevated risk of type 2 diabetes. Individuals with CHIP also had strong associations with cardiovascular conditions. Notably, CHIP was associated with elevated risk of coronary heart disease and ischemic stroke [9].

In a more recent study, compared to individuals without CHIP, carriers of CHIP variants were found to be 1.9 times as likely to have coronary heart disease, 4.0 times as likely to experience early-onset myocardial infarction, and had increased coronary artery calcification [57]. In another study, somatic *TET2* or *DNMT3A* variants in individuals with chronic ischemic heart disease appeared to associate with worse long-term clinical outcome due to heart failure. The VAF of clonal populations was associated with the severity of the clinical outcome, suggesting a dose-response relationship [58]. Similarly, aortic valve stenosis patients with *DNMT3A* or *TET2* variants who underwent aortic valve implantation surgery were observed to have heightened medium-term all-cause mortality compared to those without CHIP mutations [59]. These results were supported by the finding that mice with hematopoietic or myeloid *Tet2*-deficiency had greater cardiac dysfunction and worsened remodeling following induced heart failure [60]. Hypercholesterolemic mice engrafted with either homozygous or heterozygous *Tet2*-knockout bone marrow developed larger atherosclerotic lesions than controls, suggesting that TET2 dysfunction contributes to the development of atherosclerotic plaques [57,61]. The increased lesion size in mice that received heterozygous *Tet2*-knockout bone marrow provides evidence for haploinsufficiency as a potential mechanism for atherogenesis. This is especially important considering that most individuals with *TET2* somatic variants have only a single defective allele.

The involvement of hematological *TET2*-deficiency in the pathogenesis of atherosclerosis has been investigated at the cellular level [57,61]. *Tet2*-deficiency in myeloid cells was found to independently accelerate atherogenesis in mice and highlighted a potential role of *Tet2*-deficient macrophages as agents of atherosclerosis. Fuster et al. (2017) identified that *Tet2*-deficiency in murine macrophages facilitates nod-like receptor family pyrin domain containing 3 (NLRP3) inflammasome-dependent production of the inflammatory cytokine interleukin 1 beta (IL-1b). IL-1b increases aortic expression of endothelial adhesion markers that recruit monocytes. Importantly, *Tet2*-deficient mice that were given an NLRP3 inhibitor had reduced size of atherosclerotic lesions and reduced expression of the endothelial adhesion marker [61]. A separate study arrived at similar conclusions, showing that *Tet2*-deficient mice treated with an NLRP3 inhibitor were protected against accelerated cardiac dysfunction [60]. These findings suggest that IL-1b or NLRP3 inflammasome inhibitors may be useful therapeutic approaches in the treatment or prevention of cardiovascular disease in people with *TET2* mutations.

While somatic *TET2* changes are common in myeloid malignancy, *TET2* mutations have also been documented in the germline [62]. Interestingly, a family heterozygous for a truncating *TET2* mutation had multiple cases of lymphoma among affected individuals but no abnormal cardiovascular phenotype [63]. Monocyte-derived macrophages that were isolated from *TET2*-mutation carriers showed a hypermethylation phenotype but did not have altered cytokine or chemokine secretion. This trend was affirmed in three unrelated individuals harbouring different germline *TET2* mutations. Members of another family that were affected with a different heterozygous *TET2* germline frameshift mutation all developed myeloid malignancy, possessed thyroid abnormalities, and had no history of cardiovascular disease [64]. The apparent phenotypic contrast between those with germline vs. somatic *TET2* mutations merits further investigation with larger cohorts.

In addition to being associated with increased all-cause mortality, NGS has revealed that CHIP is significantly associated with chronic pulmonary disease [33,56]. Individuals with somatic variants in *TET2* were found to have an elevated prevalence of self-reported asthma or chronic obstructive pulmonary disease (COPD), though the nature of this relationship is unclear [65]. Future studies will need to determine whether the relationship between *TET2* mutations and COPD is mediated by smoking, or whether loss of function of TET2 plays a contributing role in the exacerbation of pulmonary diseases.

Regarding *DNMT3A*, variants in this gene in CHIP appear to be associated with the development of gastroesophageal reflux disease (GERD), whereas CHIP variants in *TET2* or with VAF > 0.1 are potentially associated with elevated levels of circulating thyroid-stimulating hormone [56], although these require further validation. In addition, our unpublished data shows that *Tet2*-deficient mice spontaneously develop pulmonary arterial hypertension associated with increased lung inflammation. It is likely that subsequent studies will reveal further comorbidities associated with CHIP. 

## 8. CHIP, Inflammation, and the Connection to the Pathogenesis of Myeloid Cancers

Many of the comorbid diseases associated with CHIP also have a shared inflammatory basis. Understanding the impact of inactivating *TET2* and *DNMT3A* mutations on myeloid cells is critical for elucidating the connection between CHIP, inflammation, and the pathogenesis of comorbid disease, as well as myeloid malignancies. The TET2 protein plays a key role in myeloid cell function as an epigenetic regulator for cell differentiation and the inflammatory response. TET2 has been identified as a mediator of transcriptional regulation for inflammatory cytokines, notably interleukin 6 (*IL6*). During the resolution of inflammation, TET2 normally recruits histone deacetylase 2 (HDAC2) to deacetylate *IL6*, thereby repressing its transcription and IL-6 levels [66]. This epigenetic change is an important regulatory step for the termination of the inflammatory state in myeloid cells, especially macrophages and dendritic cells. In vitro, murine macrophages with *Tet2* knocked out were observed to have upregulated *Il6* expression, as well as upregulation of *Il1b* and *arginase 1 (Arg1)*, during the late-phase response to lipopolysaccharides (LPS) exposure [67]. *Tet2*-knockout mice had elevated IL-6 production and were more prone to developing endotoxin shock and dextran-sulfate-sodium-induced colitis. Moreover, *TET2* loss may alter the immune environment by interfering with other types of leukocytes, as *Tet2*-deficient murine T-cells showed aberrant cytokine signaling [68]. This suggests that the loss of *Tet2* resulted in an exacerbated inflammatory phenotype, especially in myeloid cells.

It is well recognized that the prevalence of *TET2* somatic variants increases with age [9,32,33]. However, the mechanism for the emergence and subsequent dominance of *TET2*-mutated clonal populations in elderly people has not been well studied. A recent study by Abegunde et al. (2018) showed that *Tet2*-knockout murine and *TET2*-deficient human HPSCs had a proliferative advantage when chronically exposed to tumour necrosis factor alpha (TNF-α) in vitro. Upon prolonged plating with TNF-α, HSPCs with inactivating *TET2* mutations developed a resistance to apoptosis and a propensity for myeloid differentiation [69]. Similarly, upregulated *Il6* expression from HSPCs in murine *Tet2*-knockout mice (in response to acute inflammatory stress) led to apoptotic resistance via increased expression of pro-survival genes and decreased expression of pro-apoptotic genes [70]. These results suggest that the expansion of *TET2-*mutated clones may be facilitated by a resistance to an inflammatory environment that these clones help propagate (Figure 4). Furthermore, another group found that microbial signals from the intestine appear to trigger the expansion of myeloid *Tet2*-deficient clones, where bacterial translocation in *Tet2*-knockout mice led to increased IL-6 production and myeloid expansion [71]. 

Regarding malignant disease, mutant HSPC clones have been found to alter the inflammatory microenvironment associated with the bone marrow, which can facilitate clonal expansion that leads to myeloid malignancies such as MDS [72]. This modified microenvironment becomes conducive to MDS development, often resulting from increased expression of proinflammatory genes such as *IL1B*, *TNF* and *IL6*, alterations in related immune cell subsets, and susceptibility to apoptosis within the bone marrow [72]. MDS can then progress to AML; in fact, approximately 40% of MDS cases progress [73].

Increased proinflammatory mediators can be produced by the mutant HSPC clone, macrophages, dendritic cells, and T-cells, among others. Mutations in *TET2* (and other genes) can further promote proinflammatory gene expression. The resulting increase in inflammation can facilitate clonal expansion of mutant HSPCs and corresponding MDS pathogenesis or progression [72]. Macrophages, especially, have been shown to contribute to immune dysregulation and myeloid cancer pathogenesis in certain cases [74,75,76]. Other studies suggest that an abnormal immune environment in the bone marrow of patients with myeloid malignancies promotes genotoxic stress and pyroptosis, with the latter being mediated through activation of the NLRP3 inflammasome [77]. This in turn contributes to disease pathophysiology and clonal evolution that can lead to HSPC dysfunction [78,79,80].

## 9. Promises and Potential Pitfalls of Surveillance for CHIP in Human Aging

While CHIP is widely prevalent in elderly people and can pose serious health concerns through its associated risks, the question of how to effectively approach the clinical management and surveillance of CHIP is less clear. An important clinical objective for addressing CHIP should be the minimization of clonal expansions that contribute to disease states. Some have suggested that the selective destruction of mutant HSPCs is a valid strategy for the treatment of disorders originating from such HSPCs [10,13]. While this could prevent uncontrolled proliferation and subsequent disease, older individuals may rely on overly proliferative HSPCs to maintain normal blood cell levels. For such treatment to be considered, the benefit of reducing disease risk would have to be balanced with the potential health implications of treatment-induced cytopenias. Future management strategies should also address the effect of the CHIP inflammatory phenotype on broader comorbidities. *Tet2*-deficiency facilitates the upregulation of inflammatory cytokines (notably IL-1b) that contribute to the pathogenesis of atherosclerosis in mice [61]. Complementary to these findings, an IL-1b inhibitor has been shown to reduce the risk of recurrent cardiovascular events [81]. However, despite blocking IL-1b, residual IL-6 and interleukin 18 (IL-18) contribute to increased incidence of future cardiovascular events [82]. Another recent study showed that the presence of the IL-6 receptor (*IL6R*) p.Asp358Ala variant abolishes the cardiovascular risk in people with CHIP [83]. It appears that targeting the inflammatory outputs associated with dysregulated immune function may be a promising strategy for mitigating the risk of certain comorbidities in CHIP. To better understand this potential avenue of treatment, future research should seek to elucidate the mechanisms that underpin the pathogenesis of CHIP-associated diseases.

A diagnosis of CHIP can provide opportunities for management and preventative action. Hundreds of mutations in over fifty genes contribute to CHIP development [9]. These diverse mutational profiles may provide unique avenues to disease pathogenesis and an opportunity for personalized treatments. It is worth re-stating that the absolute risk for hematological cancer development in CHIP is low: 0.5% to 1.0% of individuals with CHIP develop myeloid cancer annually [9,32]. In addition, variants in *TET2* and *DNMT3A* do not significantly lower the 10-year survival rate for elderly individuals (>85-year old) [84]. However, mutations in other CHIP-related genes have been found to associate with increased risk of and accelerated time to AML development [48,85]. Importantly, models have been constructed that can predict the development of AML either months or years in advance of disease onset by analyzing the somatic molecular profiles or clinical parameters (notably red blood cell distribution width) of people with CHIP [48]. Despite this recent progress, the risk associations between CHIP variants and other comorbid diseases remain poorly understood. 

For CHIP surveillance to be implemented, it is important to consider the actionability of a CHIP diagnosis (i.e. its impact on clinical management). Identification of a pathogenic somatic variant in an otherwise hematological malignancy-free individual could warrant elevated follow-up frequencies to monitor potential progression. Individuals with other comorbidities that undergo testing for CHIP could benefit from a more informed prognosis based on the presence (or lack of) known pathogenic variants. Treatment plans for specific comorbidities, such as cardiovascular disease, could be individualized based on the calculated risk associated with testing results. However, studies concerning best practice for the treatment and prevention of CHIP-associated comorbidities have not yet been conducted, so the responsiveness of unique CHIP mutational profiles to various treatment approaches is not presently known. With CHIP clinics and related initiatives currently underway, we call for prospective studies at the involved institutions to help establish best practice recommendations.

The benefits of a CHIP diagnosis will also have to be balanced against the ramifications of screening healthy individuals. For example, using blood tests to screen for disease is invasive and risks provoking anxiety in patients. One should also consider potential ethical, legal, and financial issues (i.e. insurance eligibility) related to screening. An informed decision regarding the merit of CHIP surveillance in the general population will optimally account for all these factors.

## 10. Promise of Targeting the “Good Copy” of TET2

While several studies have assessed *Tet2* loss-of-function in mice and reported associated HSPC aberrant self-renewal and myeloid lineage expansion [35], and though screening techniques have improved greatly in recent years to detect mutations in *TET2* and other genes, therapeutic options to restore *TET2* activity are currently lacking. In a recent study, Cimmino et al. (2017) modeled restoration of *TET2* expression using reversible transgenic RNAi mice to confirm its reversal of aberrant HSPC self-renewal. Their study suggests that indirect restoration of *TET2* function may offer a new therapeutic strategy for CHIP, MDS, and AML [39]. They found that *Tet2* knockdown led to aberrant HSPC self-renewal, while Tet2 restoration reversed this effect in vitro and in vivo. Additionally, they found that *Tet2*-restored cells exhibited increased cell death, decreased proliferation, and priming toward myeloid differentiation [39]. These findings indicate that sustained *TET2* deficiency is required for disease maintenance, and that restoration of *TET2* may offer a viable therapeutic option for patients with myeloid malignancies possessing *TET2* mutations. 

Vitamin C is an α-KGDD co-factor that has been shown to promote activity of TET enzymes [86]. Administration of vitamin C has been tested, and demonstrated some efficacy, in the treatment of solid tumours [87]. To investigate the potential use of vitamin C in treating hematological malignancies, Cimmino and colleagues treated mouse HSPCs and human leukemia cells with vitamin C. They found that vitamin C treatment mimics *TET2* restoration in *Tet2*-deficient mouse HSPCs by increasing 5hmC formation, as seen in other studies [88]. Similarly, Agathocleous et al. (2017) demonstrated that ascorbate (vitamin C) regulates TET2 function in hematopoietic stem cells (HSCs). They also found that feeding ascorbate to leukemogenic mice significantly extended survival, in addition to reducing myeloblasts in the blood, spleen cellularity and HPC frequency in the spleen [89]. While these findings support the use of vitamin C to treat TET2-deficient myeloid malignancies and potentially CHIP patients with *TET2* mutations, the quantity of vitamin C required to achieve adequate TET2 level increases remains a barrier to effectively utilizing vitamin C as a treatment option for these patients. However, some more recent studies have begun to explore the use of ascorbate in a clinical setting in *TET2*-mutant patients [90,91], with one report demonstrating a clinical response of a patient with *TET2*-mutant AML to ascorbate, despite the *TET2*-mutant clone remerging at relapse [90].

Regarding targeting DNMT3A, few studies have explored the protein as a therapeutic target. While HMA treatment holds some promise for those mutations in *DNMT3A*, their toxicities remain too severe to justify use in patients with CHIP. However, some groups have begun to consider possible methods of targeting this protein. Rau et al. (2016) present a study in *Blood* that suggests that the histone lysine methyltransferase, disruptor of telomeric silencing 1-like (DOT1L) may represent a promising target in cases of *DNMT3A*-mutant AML. They demonstrated that pharmacologic inhibition of DOT1L in *DNMT3A*-mutant cell lines resulted in cell-cycle arrest and terminal differentiation, in addition to reducing cellular proliferation and inducing apoptosis in vitro [92]. Furthermore, Adema et al. (2017) published an abstract in *Blood* that demonstrated that *S*-adenosyl-l-methionine (SAM) supplementation in cases of monoallelic *DNMT3A* loss results in up-modulation of the wild-type *DNMT3A* allele in vitro [93]. These studies indicate promise in targeting levels of *TET2* and *DNMT3A* in cases of heterozygous loss-of-function mutations, though further studies are needed before these therapeutic strategies could be considered from a clinical perspective. 

## 11. Summary

Recent advances in genomic sequencing technologies have helped to characterize the phenotypic associations of acquired loss-of-function mutations in *TET2*, a gene that encodes an epigenetic regulator highly important for myeloid cell function. These mutations (along with variants in the *DNMT3A* gene) are associated with an increased risk of hematological malignancy and are especially prevalent in the elderly population. In addition, deficiency in *TET2* provokes an inflamed immune phenotype (largely via aberrant cytokine secretion) and associates with several inflammation-based diseases, with a notable causal role in cardiovascular disease. Recently, mutational status and routine clinical information has been shown to predict future development of AML in individuals with CHIP. As a common and significant risk factor, the justification for screening at-risk individuals for *TET2* mutation status must be balanced against potential ramifications of testing. While anti-inflammatory drugs can attenuate the cardiovascular risk, there are currently no viable therapeutic options for rescuing TET2 insufficiency in myeloid malignancies. Some approaches (notably vitamin C administration) have demonstrated restorative properties for TET2. However, barriers remain to these being effective treatments. Recent studies provide hope for future therapeutic strategies that may address loss-of-function mutations in both the *TET2* and *DNMT3A* genes in cases of myeloid malignancies, and potentially even CHIP and comorbid diseases. 

## Figures and Tables

**Figure 1 ijms-21-00626-f001:**
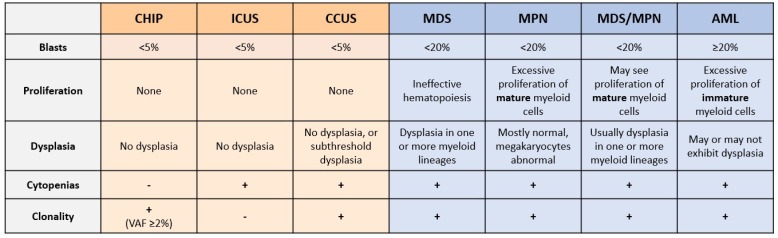
Summary of current pre-neoplastic states and myeloid neoplasms. Pre-neoplastic states (in orange: clonal hematopoiesis of indeterminate potential, CHIP; idiopathic cytopenia of undetermined significance, ICUS; and clonal cytopenias of undetermined significance; CCUS) are classified primarily based on the presence of clonality (acquired DNA variants or chromosomal aberrations), and on the presence of peripheral cytopenias [13,14]. The main categories of myeloid neoplasm (blue) are primarily classified based on blast percentage (morphologically primitive cells, including HSPCs), by the type of cells that proliferate aberrantly, and by the morphology of the myeloid cells [15]. VAF, variant allele frequency; MDS, myelodysplastic syndromes; MPN, myeloproliferative neoplasms; MDS/MPN, overlapping MDS and MPN; AML, acute myeloid leukemia.

**Figure 2 ijms-21-00626-f002:**
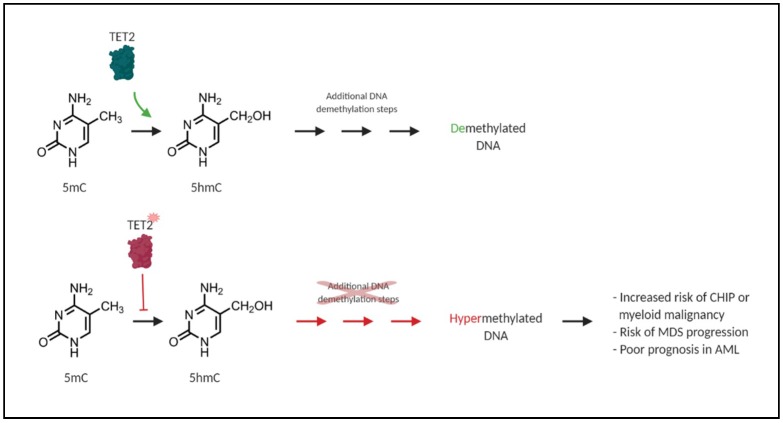
Loss-of-function mutations in *TET2* result in DNA hypermethylation. TET2 is an alpha-ketoglutarate- and Fe^2+^-dependent dioxygenase (α-KGDD) that catalyzes the oxidation of 5-methylcytosine (5mC) to 5-hydroxymethylcytosine (5hmC). This is a required step in proper DNA repair and DNA demethylation (green). Loss-of-function mutations in *TET2* in CHIP and myeloid malignancies disrupt this oxidation step and result in a general DNA hypermethylation phenotype (red) and aberrant HSPC self-renewal, which is associated with an increased risk of CHIP, myeloid malignancy, MDS progression, and poor prognosis in AML [35].

**Figure 3 ijms-21-00626-f003:**
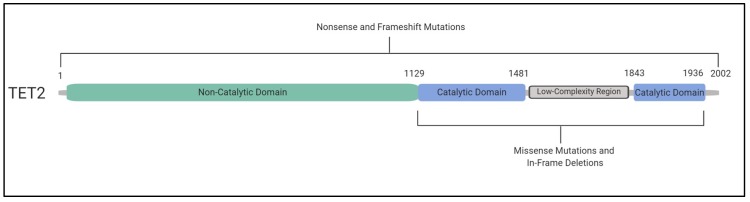
Inactivating *TET2* mutations occur throughout the coding region. The C-terminal catalytic domain of TET2 occurs approximately between amino acid residues 1129 and 1936, including a double-stranded β helix and cysteine-rich domain, binding sites for Fe(II) and α-ketoglutarate, as well as a low-complexity linker region. Together, these regions allow the catalytic domain of TET2 to bind to DNA, allowing the oxidation of 5mC to 5hmC [40]. Missense mutations and in-frame deletions occur in the catalytic domain of the protein (blue). Nonsense and frameshift mutations can occur throughout the entire coding region. However, they occur most frequently in the non-catalytic domain (green). Mutations in *TET2* are generally inactivating regardless of their nature or location in the coding region of the gene [8,40,41].

**Figure 4 ijms-21-00626-f004:**
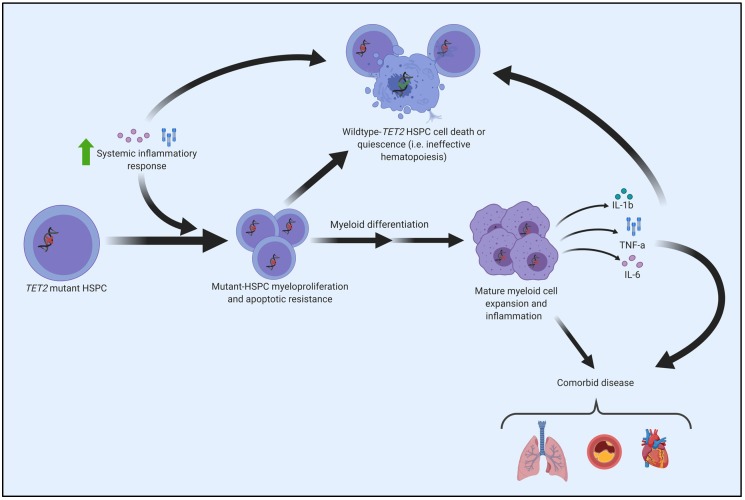
Molecular pathogenesis of *TET2*-mutated HSPC. Exposure to upregulated inflammatory cytokines (due to systemic inflammation or local alterations) advantages *TET2*-mutant HSPCs, fostering a phenotype characterized by increased proliferation and a resistance to apoptosis [69,70,71]. *TET2*-mutant HSPCs and the abnormal immune microenvironment of the bone marrow can cause premature death or quiescence of wildtype (i.e., normal) HSPCs, consequently manifesting as ineffective and clonal hematopoiesis. The increased proliferation associated with *TET2*-mutant HSPCs leads to the expansion of *TET2*-clonal populations in the bone marrow and/or peripheral blood. Mature myeloid cells (e.g., macrophages) derived from *TET2*-mutant HSPCs demonstrate a hyper-inflammatory phenotype that contributes to the pathogenesis of comorbidities, with a notable causal role in cardiovascular disease [57,61,67].

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
