# Peer review of "Age-Associated TET2 Mutations: Common Drivers of Myeloid Dysfunction, Cancer and Cardiovascular Disease"

_ijms, 2020, doi:10.3390/ijms21020626_

Round 1

Reviewer 1 Report

This is a nice review on the clinical implications of somatic TET2 mutations that drive clonal hematopoiesis. I have a few comments that I hope will help improve this manuscript:

It would be interesting to briefly discuss the difference between the biological phenomenon of clonal hematopoiesis, which can be driven by mutations and non-mutational mechanisms (as suggested for instance by Zink et al Blood 2017), and the clinical entity of CHIP, which is defined based on very specific arbitrary criteria. In line 108, it seems relevant to cite two recent papers that reported the frequency of somatic mutations in the hematopoietic system: Lee-Six et al Nature. 2018 Sep;561(7724):473-478. doi: 10.1038/s41586-018-0497-0 and Osorio et al Cell Rep. 2018 Nov 27;25(9):2308-2316.e4. doi: 10.1016/j.celrep.2018.11.014. It would be interesting to discuss briefly the nature of CHIP-related TET2 mutations, SNVs, indels… and the location of these mutations within the gene. Line 188. Associations with early-onset myocardial infarction were not evaluated in Ref.9. Line 192. For accuracy, please specify that Ref 53 evaluated the outcomes of heart failure, not chronic ischemic heart disease. Line 199. Hypercholesterolemic instead of hypercholesteremic. When discussing non-cardiovascular comorbidities potentially linked to CHIP, the authors may want to consider mentioning diabetes, as reported by Jaiswal et al, NEJM 2014.

Author Response

RESPONSE TO REVIEWERS (responses in blue text)

Manuscript ID: ijms-684798

Title: Age-associated TET2 mutations: common drivers of myeloid dysfunction, cancer and cardiovascular disease

Authors: Christina K. Ferrone, Mackenzie Blydt-Hansen, Michael J. Rauh *

Received: 17 December 2019

Submitted to section: Molecular Immunology, Myeloid Cell Heterogeneity in Health and Disease

Comment from Editor:

It has been reviewed by experts in the field and we request that you make minor revisions before it is processed further.  Please revise the manuscript according to the reviewers' comments and upload the revised file within 5 days. Use the version of your manuscript found at the above link for your revisions, as the editorial office may have made formatting changes to your original submission. Any revisions should be clearly highlighted, for example using the "Track Changes" function in Microsoft Word, so that they are easily visible to the editors and reviewers. Please provide a short cover letter detailing any changes, for the benefit of the editors and reviewers.

Reviewer 1 – Comments and Suggestions for Authors:

This is a nice review on the clinical implications of somatic TET2 mutations that drive clonal hematopoiesis. I have a few comments that I hope will help improve this manuscript:

We sincerely thank Reviewer 1 for taking the time to review our manuscript and providing helpful suggestions to improve its quality.

It would be interesting to briefly discuss the difference between the biological phenomenon of clonal hematopoiesis, which can be driven by mutations and non-mutational mechanisms (as suggested for instance by Zink et al Blood 2017), and the clinical entity of CHIP, which is defined based on very specific arbitrary criteria.

We thank Reviewer 1 for this suggestion and apologize for not being clearer about these definitions and biological differences between types of clonal hematopoiesis.  In the first paragraph of Section 5, “CHIP and risk of hematological malignancy”, we have added the following text:

“Furthermore, while CHIP is a clinical entity with specific criteria, including mutations in recognized cancer driver genes like TET2 and DNMT3A, the more general term clonal hematopoiesis (CH) describes a state in which a single hematopoietic stem cell clone gives rise to a disproportionate number of an individual’s mature blood cells. CH can include mutations at VAF <2% but may also arise without recognizable cancer driver mutations, possibly resulting from contraction and neutral drift in the HSPC pool.”

In line 108, it seems relevant to cite two recent papers that reported the frequency of somatic mutations in the hematopoietic system: Lee-Six et al Nature. 2018 Sep;561(7724):473-478. doi: 10.1038/s41586-018-0497-0 and Osorio et al Cell Rep. 2018 Nov 27;25(9):2308-2316.e4. doi: 10.1016/j.celrep.2018.11.014.

We thank the Reviewer for pointing out these pertinent references.  We now cite them in this revised passage that starts on line 108 (they are assigned reference numbers 11 and 30):

“Hematopoietic cells tend to accumulate somatic mutations over time. It has been estimated that there are normally 50,000 – 200,000 HSPCs, and that they have a fidelity rate of about 0.78 × 10−9 mutations per genomic base pair per cell division, resulting in random mutations occurring at a rate of approximately 14 base substitutions and 0.13 coding mutations per year of life (i.e. approx. one mutation every 7-8 years)[7], [11], [30].”

It would be interesting to discuss briefly the nature of CHIP-related TET2 mutations, SNVs, indels… and the location of these mutations within the gene.

We agree and thank the Reviewer for this suggestion.  We include in the revised manuscript an additional figure (new Figure 3) illustrating and describing the structure of the CHIP protein, and the location and impact of mutations with respect to TET2 function.  We also include a new passage starting on revised line 160 that discusses the nature of CHIP-related SNVs and indels:

“Several studies have assessed the structure of the TET2 protein and the context of somatic mutations in myeloid malignancies, including the domains in which they occur. Whether mutations occur prior to or within the C-terminal catalytic domain of TET2, they generally result in loss of enzymatic function. Studies have shown that most commonly, nonsense or frameshift mutations occur before (and occasionally within) the catalytic domain, while missense mutations and in-frame deletions occur within the catalytic domain[8], [40], [41] (Figure 3). To our knowledge, a comprehensive comparison of TET2 mutations in CHIP versus myeloid malignancies has yet to be performed; however, the nature of TET2 CHIP mutations are also in keeping with TET2 catalytic inactivation.”

Line 188. Associations with early-onset myocardial infarction were not evaluated in Ref.9.

Line 192. For accuracy, please specify that Ref 53 evaluated the outcomes of heart failure, not chronic ischemic heart disease. Line 199. Hypercholesterolemic instead of hypercholesteremic. When discussing non-cardiovascular comorbidities potentially linked to CHIP, the authors may want to consider mentioning diabetes, as reported by Jaiswal et al, NEJM 2014.

We apologize for any inaccuracies and thank Reviewer 1 for catching these. The text has been revised accordingly between lines 220 – 242 in the new manuscript version.

Reviewer 2 – Comments and Suggestions for Authors:

Christina K. Ferrone, Mackenzie Blydt-Hansen and Michael J. Rauh present a review about TET2 mutations and discuss recent findings in hematopoiesis and hematological diseases. The manuscript focuses particularly on age-related clonal hematopoiesis and its close relationship to disease progression to myeloid malignancies but also inflammatory and cardiovascular diseases. The issues raised in the final sections are of particular interest in the current era of medicine. The manuscript is clear and well written. I have only a few suggestions for discussion, especially regarding the non-hematological aspects of TET2-mediated pathogenesis.

We sincerely thank Reviewer 2 for taking the time to review our manuscript and providing helpful suggestions to improve its quality.

At least 3 groups have reported germline/inherited TET2 mutations (Schaub FX, et al. Blood. 2010;115(10):2003-2007. doi:10.1182/blood-2009-09-245381 ; Kaasinen E, et al. Nat Commun. 2019;10(1):1252. doi:10.1038/s41467-019-09198-7 ; Duployez N, et al. Leukemia. December 2019. doi:10.1038/s41375-019-0675-6). Germline TET2-mutated individuals present with a wide range of myeloid but also lymphoid malignancies. Intriguingly, none of them present with a predisposition to atherosclerosis or abnormal pro-inflammatory cytokine expression. The authors should discuss these findings.

Indeed, the reports of these 3 groups are intriguing and we thank the Reviewer for this suggestion.  We hope the Reviewer will find the following additional passage satisfactory, which appears on lines 261 – 271 of the revised manuscript, and includes the new references:

“While somatic TET2 changes are common in myeloid malignancy, TET2 mutations have also been documented in the germline[62]. Interestingly, a family heterozygous for a truncating TET2 mutation had multiple cases of lymphoma among affected individuals but no abnormal cardiovascular phenotype[63]. Monocyte-derived macrophages that were isolated from TET2-mutation carriers showed a hypermethylation phenotype but did not have altered cytokine or chemokine secretion. This trend was affirmed in three unrelated individuals harbouring different germline TET2 mutations. Members of another family that were affected with a different heterozygous TET2 germline frameshift mutation all developed myeloid malignancy, possessed thyroid abnormalities, and had no history of cardiovascular disease[64]. The apparent phenotypic contrast between those with germline vs. somatic TET2 mutations merits further investigation with larger cohorts.”

When discussing ascorbate as a potential treatment in TET2-mutated patients, the authors could cite the recent report by Das AB et al (Blood Cancer Journal. 2019;9(10). doi:10.1038/s41408-019-0242-4) describing a patient with clinical response to ascorbate (albeit temporary).

This is an excellent suggestion and we thank Reviewer 2.  We now include this practical example of a clinical response to ascorbate (new ref. 91) in the revised text, along with another example we subsequently found (new ref. 90) between lines 437 – 440 of the revised text:

“However, some more recent studies have begun to explore the use of ascorbate in a clinical setting in TET2-mutant patients[90], [91], with one report demonstrating a clinical response of a patient with TET2-mutant AML to ascorbate, despite the TET2-mutant clone remerging at relapse[90].”

Minor point: I recommend removing the word "cytoses" from Figure 1 since this word is not used in literature and may be confusing with other terms related to cell biology (exocytosis, endocytosis, phacocytosis).

We appreciate the suggestion to avoid confusing terminology and we have removed “cytoses” from Figure 1.

Reviewer 2 Report

Christina K. Ferrone, Mackenzie Blydt-Hansen and Michael J. Rauh present a review about TET2 mutations and discuss recent findings in hematopoiesis and hematological diseases. The manuscript focuses particularly on age-related clonal hematopoiesis and its close relationship to disease progression to myeloid malignancies but also inflammatory and cardiovascular diseases. The issues raised in the final sections are of particular interest in the current era of medicine. The manuscript is clear and well written.

I have only a few suggestions for discussion, especially regarding the non-hematological aspects of TET2-mediated pathogenesis.

At least 3 groups have reported germline/inherited TET2 mutations (Schaub FX, et al. Blood. 2010;115(10):2003-2007. doi:10.1182/blood-2009-09-245381 ; Kaasinen E, et al. Nat Commun. 2019;10(1):1252. doi:10.1038/s41467-019-09198-7 ; Duployez N, et al. Leukemia. December 2019. doi:10.1038/s41375-019-0675-6). Germline TET2-mutated individuals present with a wide range of myeloid but also lymphoid malignancies. Intriguingly, none of them present with a predisposition to atherosclerosis or abnormal pro-inflammatory cytokine expression. The authors should discuss these findings.

When discussing ascorbate as a potential treatment in TET2-mutated patients, the authors could cite the recent report by Das AB et al (Blood Cancer Journal. 2019;9(10). doi:10.1038/s41408-019-0242-4) describing a patient with clinical response to ascorbate (albeit temporary).

Minor point: I recommend removing the word "cytoses" from Figure 1 since this word is not used in literature and may be confusing with other terms related to cell biology (exocytosis, endocytosis, phacocytosis).

Author Response

(The authors gave the same response as above.)
